# Knowledge, Attitude and Practices towards the Tiger Mosquito Aedes Albopictus. A Questionnaire Based Survey in Lazio Region (Italy) before the 2017 Chikungunya Outbreak

**DOI:** 10.3390/ijerph17113960

**Published:** 2020-06-03

**Authors:** Beniamino Caputo, Mattia Manica, Gianluca Russo, Angelo Solimini

**Affiliations:** 1Department of Public Health & Infectious Diseases, University of Roma “La Sapienza”, 00185 Rome, Italy; beniamino.caputo@uniroma1.it (B.C.); gianluca.russo@uniroma1.it (G.R.); 2Department of Biodiversity and Molecular Ecology, Research and Innovation Centre, Fondazione Edmund Mach, 38010 San Michele all’Adige, Italy; mattia.manica@fmach.it

**Keywords:** KAP mosquito, knowledge, awareness and practice, Aedes-borne diseases, Aedes albopictus, mosquito, mosquito life cycle, arboviruses, infectious disease

## Abstract

The invasion of *Aedes albopictus* has played a major role in the resurgence of mosquito-borne diseases in Italy, generating the two largest chikungunya outbreaks in Europe (2007, 2017). Knowledge, attitude and practice (KAP) are important in order to prevent *Aedes*-borne disease transmission, yet so far they have not been assessed. To this scope we used multivariate logistic regression to investigate KAP of citizen-to-*Aedes* ecology and transmitted diseases. Data were collated by a structured questionnaire (18 questions) in 2016. Participants were selected in the Lazio region from members of native populations and two resident communities (RC) originating from the Indian subcontinent where *Aedes*-transmitted diseases are endemic. Results showed that compared to Italians, RC respondents had a higher knowledge and concern of *Aedes*-transmitted diseases (Odds Ratio = 2.61 (95%CI: 1.03–6.05); OR = 3.13 (2.15–4.65)) as well as their life cycles (OR = 2.49 (1.75–3.56); OR = 9.04 (6.22–13.66)). In contrast, they perceived a lower nuisance due to the presence of *Ae. albopictus* (OR = 0.2 (0.13–0.32); OR = 0.55 (0.38–0.78). These findings suggest that citizens in the Lazio region are not prepared to face a potential outbreak of arboviruses and further efforts should be made to increase knowledge, awareness and best practices.

## 1. Introduction

The mosquito-borne disease burden is increasing worldwide following globalization and the expansion of travel and trade from areas colonized by vectors and pathogens. A good example of the association between global human travel and mosquito invasions into new areas is represented by *Aedes albopictus*. The so-called “tiger-mosquito” is one of 100 of the most invasive animal species in the word and in fewer than 30 years it has spread across the five continents, colonizing large areas [1]. The rapid expansion of this species was caused by the worldwide trade of tires and the ability of the tiger mosquitoes to produce eggs that diapause and resist the relatively cold winters of temperate areas [2,3]. 

In non-endemic countries like Italy, the spread of arboviruses is facilitated by the flux of travelers that included sick individuals (i.e., viremic) arriving from endemic areas, increasing the risk of local outbreaks of dengue, chikungunya and yellow fever [4,5,6]. Predictive models [6] and surveillance data suggest that the majority of imported cases involve Italian residents travelling in tropical areas for holidays or business or members from long established communities of migrants returning to Italy after visiting friends and relatives in their home country, as it was with the case of the chikungunya outbreak in 2007 [7]. One of the key factors in preventing the expansion of local outbreaks obviously includes the public health system preparedness to act and respond once early cases have been clinically discovered [8]. However, most clinical symptoms of chikungunya are non-specific and these diseases are generally self-limiting with no need for hospitalization [9]. Additionally, awareness of general practitioners in temperate regions to these rare diseases is generally low [10]. Therefore, an early detection of cases is often challenging, as demonstrated by the 2017 outbreak of chikungunya in Anzio, Italy [5,11]. Public awareness and knowledge of *Aedes*-transmitted diseases may increase the likelihood of patients being referred to a doctor if symptoms are compatible with an arboviral disease developed soon after returning from a country where the disease is endemic.

Another important factor in the prevention of *Aedes*-transmitted diseases concerns the engagement of the community [12,13]. Knowledge of the mosquito life cycle and an ability to recognize typical breeding sites of *Aedes* mosquitoes by citizens is an important factor in promoting community engagement and preventing disease transmission [12,14]. Indeed, there is some evidence on how informative campaigns to citizens could improve *Aedes* control by reducing man-made breeding sites [15,16]. Italy was among the first European countries to be invaded by *Ae. albopictus* and now all major urban areas have a well-established population, but there is a lack of data on the level of knowledge, awareness and practices (KAP) of citizens concerning *Aedes* ecology and transmitted diseases. 

Additionally, there are no data on KAP for the Asian resident communities that may be more exposed due to frequent travelling and contacts in Southeast Asia where mosquito-borne diseases are endemic.

The aim of this study was to compare levels of KAP towards *Ae. albopictus* nuisance and transmitted diseases in three groups of the general population resident in the Lazio region of Central Italy. The reference group was Italians while the other two groups were Malayalis, originally from Kerala in India; and Punjabis, originally from North India. Since in India *Aedes*-transmitted diseases are endemic and frequent outbreaks are recorded [17], a higher level of knowledge and awareness was expected from the Punjabis and the Malayalis compared to the Italians (Lazio region) where the first Chikungunya outbreak was recorded only in 2017, after the completion of this study [18].

## 2. Materials and Methods 

A descriptive, cross-sectional study was conducted using a structured questionnaire. Non-native participants were selected among members of two different communities originating from the Indian subcontinent: a community from Kerala (South India; from now on Malayalis) and their members that live in the urban area of Rome; and another community that originates from the Punjab region (North India; from now on Punjabis) and their members who live in a rural area in the province of Latina (South Lazio), not far from Anzio where the first Chikungunya outbreak occurred in 2017. Malayalis are mostly employed in the tertiary sector (mainly as personal home care givers and restaurant food processers) while the Punjabis mostly work in the agricultural sector, manly as fruit and vegetable harvesters in outdoor settings. 

Anonymous interviews were conducted face to face by trained students on several occasions during summer 2016 (August-September) on a convenient sample recruited at metro stations, train stations, churches and other meeting places where people are known to meet regularly. The majority of Malayalis and Punjabis speak and understand the Italian language and/or the English language well and interviews were conducted using the language that the responder felt most comfortable with. We rejected individuals that were not resident in the Lazio region, were in Italy less than 1 year or not speaking neither Italian nor English. Confidentiality was assured by anonymity and the absence of sensitive information. The responders had the right to refuse to participate, to refuse to answer any questions they wished and to withdraw from the survey at any moment. 

The survey instrument included a total of 18 questions on socio-demographics and housing characteristics (N = 10), knowledge (N = 3), attitudes (N = 2) and practices (N = 1) regarding the biology of *Ae. Albopictus* and *Aedes*-borne diseases. The KAP questions and their classifications followed the approach of Potter et al [19]. Whenever referring to *Ae. Albopictus* in the questionnaire/interviews, we used the colloquial term “tiger-mosquitoes”. Socio-demographic information included the municipality (or neighborhood of Rome) of residence, age, sex, years lived in Italy, educational level, the type of current housing (if in an apartment within a condominium or an independent house), house floor (ground floor or lower, first or second floor, third floor or higher), house window screen (yes or no), house outer space (balcony, garden, none) and green area nearby house (within 500 m; yes or no). 

The knowledge section included 3 questions to infer knowledge about the tiger mosquito’s life cycle, biting times and transmitted diseases. We asked for the preferred place of egg deposition and larvae development of tiger mosquitos (bare ground, walls, trees, shrubs and herbs, small water containers, storm drains, don’t know), the preferred tiger mosquitoe biting time (early morning, late afternoon, evening, night time, don’t know) and the diseases that they can transmit (don’t know, malaria, dengue or chikungunya or yellow fever, HIV, Influenza). 

In the attitude section, we asked for the perceived nuisance of mosquito presence (“How often do you feel disturbed at home by tiger mosquitoes?”) and about any irritation due to mosquito bites (“How much do you feel disturbed by tiger mosquitoes bites?”) using a 3-points scale (1 = not at all, 2 = neutral/somehow, 3 = a lot/extremely). Concern about *Aedes*-transmitted diseases was addressed with a 5-points Likert scale (“From 1 = not at all to 5= extremely worried, how often are you worried about tiger mosquitoes transmitted diseases?” and coded as 1 = not at all, 2 = a little worried, 3 = neutral/sometimes worried, 4 = very worried, 5 = extremely worried). 

In the practice section, we asked about the actions performed by responders to avoid *Aedes* bites (“What do you do to avoid tiger mosquitoes bites?”) with an open question and classified the answers during data entry as none, personal or home (environmental) repellents. This study was approved by the Department board of Public Health and Infectious Diseases in 9 October 2014 (CDDSPMI-100914). Permission to conduct the study was also obtained from the heads of the Malayalis and Punjabis communities. Confidentiality and anonymity were explained to each participant, and oral consent was obtained. 

### Data Analysis

Descriptive statistics such as frequency, mean and standard deviation (SD) were applied to assess all variables in the current study. Statistical significance for comparison of survey responses was calculated using a Chi-squared test for categorical data and an ANOVA test for continuous variables, as appropriate. To assess differences in each KAP answer, we used logistic regression models. Original categorical variables were transformed into binary variables as described below. 

A knowledge of *Ae. Albopictus*’ life cycle (multiple choice allowed for this question), biting time and transmitted diseases were recoded as 1 (correct answer for cycle: small water containers or storm drains without other options; biting: early morning or late afternoon; diseases: dengue or chikungunya or yellow fever) if answers were correct and 0 otherwise. 

Attitudes on the frequency of nuisance due to *Ae. Albopictus* presence and from irritation due to mosquito bites were recoded as 1 if “very often/always” (presence) “a lot/extremely” (bites) and, 0 otherwise. Concern about *Aedes*-transmitted diseases was coded as 1 (very often/always concerned) and 0 otherwise. 

Finally, practices to avoid *Ae. albopictus* bites were coded as 0 (none) or 1 (otherwise). To assess KAP differences from the autochthonous population (ref = Italians), we reported the effect estimates as an odds ratio (OR) and corresponding 95% confidence intervals. For each KAP binomial variable, a model was fitted that included age, sex and education level. 

We also fitted a second model which included the house characteristics just for the KAP binomial variable regarding the perceived nuisance from tiger mosquito presence at home. All analyses were carried out with R version 3.6 [20]. 

## 3. Results

### 3.1. Socio-Demographics and House Characteristics

Overall, 1579 responders agreed to participate in the study. In total we interviewed 470 individuals from India (of which 204 were Malayalis and 266 were Punjabis) and 1109 Italians. Socio-demographics and housing characteristics are reported in Table 1. The average age of respondents varied from 31 (Punjabis) to 37 (Malayalis) to 39 years (Italians). On average, respondents from India had been living in Italy for an average of 7 years (Malayalis: mean = 7, min = 1, max = 34 years; Punjabis: mean = 7.4, min = 1, max = 24 years). There were sex differences (Pearson’s Chi-squared test, Chi-squared = 277.7, df = 2, *p-*value < 0.001), as Italian respondents included slightly more females (57.2%), Malayalis slightly more men (61%) and Punjabis only men (100%). Moreover, educational levels were different, as Punjabi respondents with a high school degree or higher (32%) were fewer than Italian (78%) and Malayalis (86%) with the same (Pearson’s Chi-squared test, Chi-squared = 219.8, df = 2, *p-*value < 0.001). 

The three groups differed in house types (Pearson’s Chi-squared test, Chi-squared = 103.9, df = 2, *p-*value < 0.001), outdoor space (Pearson’s Chi-squared test, Chi-squared = 109.4, df = 4, *p-*value < 0.001), floor (Pearson’s Chi-squared test, Chi-squared = 40.1, df = 4, *p-*value < 0.001), window screen (Pearson’s Chi-squared test, Chi-squared = 245.2, df = 2, *p-*value < 0.001). Most of the apartments/houses were on the third floor (Malayalis 66%, Punjabis 89%, Italians 76%). About half (51%) of the Punjabis lived in independent houses without a house window screen (89%) and often with an outdoor garden (42%) and with a green area nearby (100%), while most Malayalis (90%) lived in apartments (with a house window screen, 84%) in urban areas without an outdoor space (balcony or garden, 52%) and without green areas nearby (64%). Two-thirds of the Italian group lived in apartments in urban areas with a house window screen (52%), a balcony (44%) or a garden (40%), with a green area nearby (74%), Table 1. 

### 3.2. KAP

The frequency of answers to the knowledge, attitude and practice questions for the three groups is reported in Appendix A and in Figure 1 and Figure 2. 

Malayalis exhibited the highest correct knowledge (72.5%) on places where tiger mosquitoes lay eggs and larvae develop (answer only “small water containers or storm drains” without other possible wrong choices such as “bare ground, vegetation, walls, don’t know”) compared to Punjabis (47%) and Italians (21.3%). A high percentage (Figure 1) of Italians correctly identified water containers or storm drains as suitable breeding sites, but also wrongly added other places, as non-exclusive multiple choices were allowed for this question. 

Malayalis exhibited a higher knowledge of biting times (early morning or late afternoon, 34.1%) and on the correct diseases transmitted (dengue, chikungunya, yellow fever, 28.7%) compared to both Punjabis and Italians. In the attitude section, Italians were the less worried by diseases that were possibly transmitted by tiger mosquitoes (15.2%), but were most disturbed by bites (58.3%) and by mosquitoes’ presence in the home (46.2%). Regarding action taken to avoid tiger mosquito bites, more than half of the Italians and Malayalis said they did nothing (Italians: 53%, Malayalis: 57.6%) compared to only one third of Punjabis: 33.8%), Figure 2.

### 3.3. Multivariate Analysis

The results of the multivariate analysis of KAP is summarized in Table 2 while Figure 3 shows predictors of perceived nuisance from tiger mosquito presence at home. 

Compared to the Italian respondents (reference group), Malayalis and Punjabis had a greater knowledge of tiger mosquitoes’ life cycles (df = 2, Deviance = 154.6, *p-*value = <0.001), biting times (df = 2, Deviance = 6.6, *p-*value = <0.036) and transmitted diseases (df = 2, Deviance = 78.2, *p-*value = <0.001). For the attitude section, Malayalis and Punjabis had a greater concern compared to the Italians about diseases transmitted by tiger mosquitoes (df = 2, Deviance = 17.9, *p-*value = <0.001) but a lower perceived nuisance of the presence (df = 2, Deviance = 60.8, *p-*value = <0.001) and bites (df = 2, Deviance = 247.2, *p-*value = <0.001) of tiger mosquitoes. The Malayalis group was not statistically different from the Italian group concerning the actions to avoid tiger mosquitoes bites, while the Punjabi group was more active in avoiding *Aedes* bites (df = 2, Deviance = 11.2, *p-*value = 0.004). After adjusting for house characteristics, model results showed that the Italian group felt more annoyed than Punjabis but not Malayalis. Moreover, perceived nuisance at home was positively correlated with the sex and age of the respondents, with woman and younger respondents reporting higher nuisance. Finally, the house characteristics that were greatly associated with perceived nuisance were those that included ground floor living and the presence of a balcony (Figure 3).

## 4. Discussion

In Italian residential areas, *Ae. albopictus* infestations are not effectively managed by conventional adulticide and/or larvicides treatments. It is hypothesized that this is due to high costs that are not affordable for public health authorities outside of emergency periods (i.e., outbreak management). For this reason, community engagement plays a key role in the prevention of vector-borne diseases [21,22] by making citizens aware of how to reduce breeding sites [15,23], which may be a good cost-effective way of reducing infestations [24]. However, the successful management of *Ae. albopictus* infestations likely depends on the KAP of residents [25]. To our knowledge, our study was the first to collect information on citizen KAP since the establishment of *Ae. albopictus* in the Lazio region and before the first chikungunya outbreak of 2017. The comparison among Italian, Malayali and Punjabi communities permitted us to highlight differences in *i)* knowledge of vector life cycles, biting behaviors and transmitted diseases; and *ii)* perceptions of tiger mosquito bite nuisances and concerns about mosquito-borne diseases. These findings could be exploited in future interventions to increase community awareness and increase vector control effectiveness in urban settings. 

Compared to the other two residence’s groups from India, the knowledge of Italian respondents was lower on specific aspects of the *Ae. albopictus* life cycle, biting activity and potentially transmitted diseases. Therefore, a lack of knowledge may not arguably be due to a lack of exposure and annoyance by *Aedes albopictus* mosquitoes. Italy was one of the first European countries to be invaded by these mosquitoes almost 30 years ago (first recorded in Genoa 1990~) [26]. Moreover, after its invasion, the tiger mosquito rapidly colonized Italy [27] and is strongly perceived as a pest and a nuisance due to different ecological and physiological factors. Firstly, in large urban areas of several regions, citizens experienced for the first time hematophagous insect bites while outdoors during day time, likely reducing the quality of the life outdoors [28]. Secondly, the tiger mosquito’s saliva was reported to be an irritant to new naive hosts [29], producing a large inflammatory response that during the process of invasion was particular relevant. Moreover, *Ae. albopictus* has a daily biting activity, corresponding to the human host activity outdoors (early morning, and late afternoon) [30]. In order to escape from the host during the bites, the tiger mosquito has selected an opportunistic behavior and bites multiple times (even on the same host) to complete a single gonotrophic stage. This biting behavior is likely to be perceived as a strong annoyance [31], even when the mosquito density is low, as probably also happened at the beginning of its colonization phase. 

The low citizen knowledge on the necessity of water for the mosquito larvae life cycle is particularly relevant from a vector control perspective. *Aedes* container mosquitoes are synanthropic—they breed in small larval sites close to human activities and humans represent their main feeding target [32]. A key component of integrated vector management of *Aedes* container mosquitoes is the engagement of citizens, which should reduce potential breeding sites in private areas. Citizens’ knowledge of the mosquito life cycle is essential to achieve this goal. However, it is possible that due to the lack of basic entomological information, most respondents confuse potential larval breeding sites with adult habitats, where they probably receive more bites (such as in nearby vegetation). Finally, the differences among the Italian and Indian communities has highlighted a huge gap that needs to be filled when starting a community engagement project to control *Aedes* mosquitoes.

While Malayalis and Punjabis scored highly in the knowledge of tiger mosquito biology and in worries of transmitted disease, the main concern of Italian respondents seems to be related to annoyance from mosquito infestations and mosquito bites rather than the potential health risks involved. This difference may be at least partially explained by different levels of irritation and itching due to *Aedes* bites, that was also higher in the Italian group. However, a higher perceived nuisance to mosquito presence could also reflect a higher exposure to *Aedes* infestations that are heterogeneously spatially distributed [33]. Although a recent study did not find an association between perceived nuisance and actual egg mosquito exposure in the south of France and in La Martinique [34], in another study conducted at La Reunion after the outbreak of Chikungunya occurred [35], the perceived tiger mosquito nuisance was linked to their role in transmitting disease by 80% of respondents.

The higher Italian perception of nuisance from tiger mosquito presence and their bites do not seem to correspond to an increase in action taken to avoid *Aedes* bites compared to those adopted by Punjabis. This may be due to the specific outdoor work environment of many Punjabis and their consequent frequent exposure to mosquito bites. Although they feel less disturbed by the presence of mosquitoes at home, Punjabis nevertheless adopted different actions to avoid *Aedes* bites. Interestingly, Punjabi respondents who worked outdoors showed a higher prevalence of use of personal repellent to avoid tiger mosquito bites at home compared to the other two groups. There are few data in Italy trying to quantify the actions and the private cost to protect people from mosquito bites. Notably in Italy, while mosquito control is coordinated at level of public areas by public tenders, in private areas vector control is carried out by different pest control companies or by private citizens without area-wide planning. The KAP results also highlighted other risk factors associated with mosquito nuisance such as living on the ground floor or having a balcony. These are likely due to the fact that balconies in urban areas represent opportunities of breeding sites for *Ae. albopictus* and are the main sources of water in urban areas during the summertime due to exposed flowers. 

The Italian population may be more vulnerable to the potential transmission of arbovirus by *Ae. albopictus* compared to other investigated groups. Indeed, the Italian citizens who participated not only had a poorer knowledge of potential breeding sites, and were likely to inadvertently breed mosquitoes in their home (e.g., on their balcony or in their gardens), but also confused the hours of exposure to *Aedes* bites with exposure to other species (e.g., *Culex pipiens*) that bite during the evening or at night. Finally, Italians showed a lower concern and a poorer knowledge of the potential role of *Ae. albopictus* as a vector of important human diseases, such as chikungunya, dengue and the zika virus (infectious diseases that are increasing worldwide) [36]. It is relevant to point out that this KAP questionnaire has been carried out nine years after the chikungunya outbreak in Emilia Romagna in 2007 [7] and that compared to India, *Aedes*-transmitted diseases are not endemic. Moreover, at the time that the questionnaire was administered, the chikungunya 2017 outbreak had not yet happened, while cases of dengue and chikungunya were still common in the Indian sub-continent at the time [17,37]. This may explain the main concern regarding nuisance rather than pathogen transmission in the Italian group. However, the lack of basic knowledge concerning *Aedes*-transmitted diseases combined with a low perception of the potential risks of *Aedes*-transmitted arbovirus is coherent with what has already been observed in the same region (Lazio) by a survey targeting general practitioners (GPs) [10]. It is difficult to speculate on the impact of the lack of citizens’ knowledge (and attitude) on tiger mosquitoes and related diseases during the last chikungunya outbreak. However, it is important to repeat a KAP questionnaire in the future in order to understand whether the last chikungunya outbreak (2017) has raised a major awareness among GPs as well as in citizens. 

*Aedes*-transmitted diseases in European temperate regions are still epidemic events dependent on imported cases and until now, outbreaks have been limited and rare [6]. The limited number of *Aedes* outbreak events combined with the mild symptoms of chikungunya diseases [11,38] and the low likelihood of hospitalization has probably not increased awareness of *Aedes*-transmitted diseases in the general population. It is therefore unlikely to expect a greater knowledge, attitude and practice around *Aedes*-transmitted diseases due to the last 2017 chikungunya outbreak. The best way to increase the awareness of citizens as well as GPs would be to organize information campaigns to educate citizens at any level of education (school, university, medical specialization) by a specific task force of experts in each Italian region that follows the last arbovirus recommendations from the Italian Minister of Health [39].

This study has several limitations that need to be addressed. Firstly, this study is based on a convenient sample and it might not be representative of the whole population, especially where the exposure to *Ae. albopictus* infestations is concerned. However, interviews on Italian citizens were collected over the whole area of Rome in order to minimize selection bias. Secondly, in order to make interviews as fast as possible for people waiting for trains in underground stations, it was kept short and with very few items per KAP domain. Additionally, the questionnaire that was used is not a validated instrument and its psychometric properties should be investigated carefully in future applications. Thirdly, respondents included in this study might have included people that are more worried about mosquito infestations than non-respondents, although only a few people refused to answer. 

## 5. Conclusions

Data collected in this survey draw a worrying picture of the vulnerability of citizens to *Ae. albopictus*. Comparing the three communities helped to highlight some critical differences in perceptions regarding coexistence with mosquitoes. The results suggest a general lack of knowledge on *Ae. Albopictus*’ life cycle, in particular its breeding sites that may directly affect a source reduction campaign and reduce vector control effectiveness. In the Italian group, *Ae. albopictus* was still perceived as a nuisance pest rather than a potential vector of diseases, thus likely leading to an underestimation of the importance of vector control activities. In the face of potential future outbreaks of arboviruses, these findings underline the urgent need to increase citizens’ knowledge and awareness as well as the dissemination of best practices to reduce *Ae. albopictus* infestation. Organizing local groups of experts, as advised by the latest recommendations from the Italian Minister of Health, with the task of improving awareness in citizens and GPs is a first step towards a mitigation of the current situation.

## Figures and Tables

**Figure 1 ijerph-17-03960-f001:**
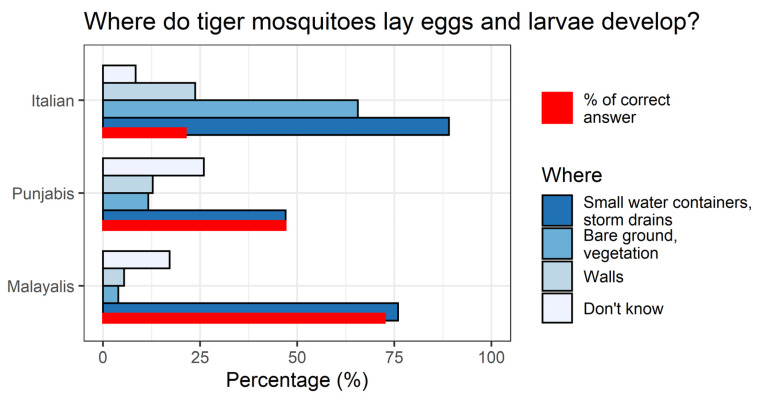
Answers regarding tiger mosquito breeding site knowledge on places where tiger mosquitoes lay eggs and larvae develop. Non-exclusive multiple answers were allowed.

**Figure 2 ijerph-17-03960-f002:**
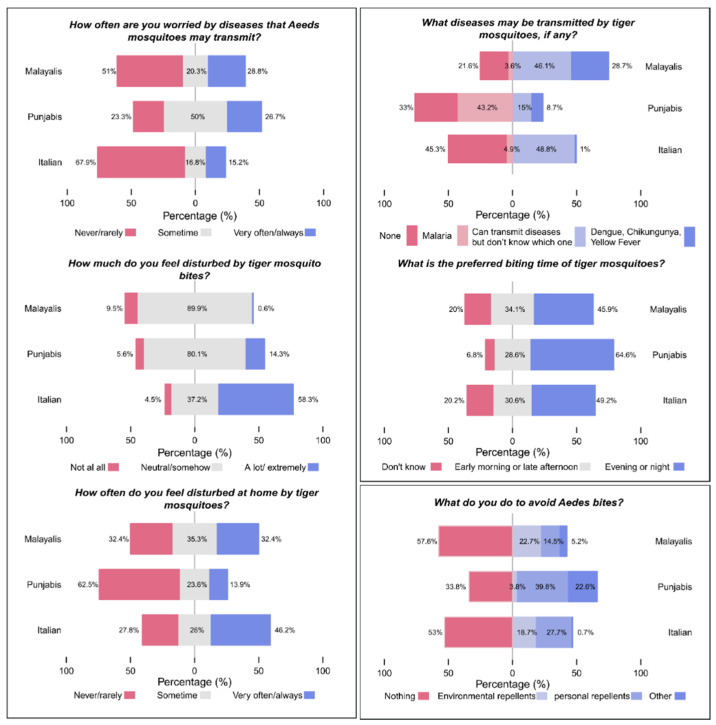
Frequency of answers to knowledge (2 out 3 questions, upper right), attitude (3 questions, left box) and practice questions (one question, low right) for the three groups (Malayalis N = 204, Punjabi N = 266, Italian N = 1109).

**Figure 3 ijerph-17-03960-f003:**
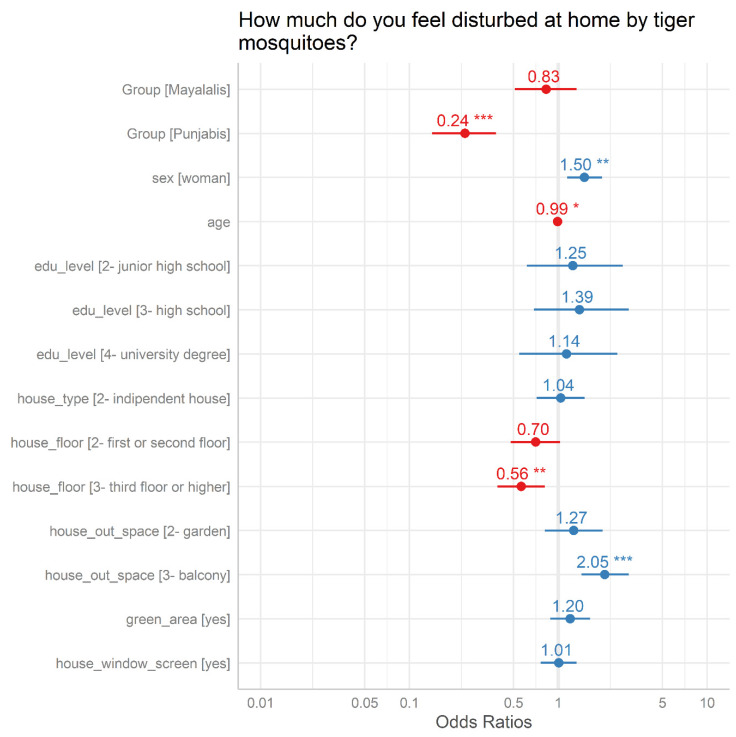
Estimates of the logistic regression model analyzing attitudes towards perceived mosquito nuisance at home. The points represent the average ORs (horizontal line: 95% confidence interval) of KAP.

**Table 1 ijerph-17-03960-t001:** Descriptions of socio-demographics and house characteristics in the three groups.

Variable		Malayalis (N = 204)	Punjabis (N = 266)	Italians (N = 1109)
Age	mean	37.2	31.3	39.6
	min	18.0	18.0	18.0
	max	69.0	52.0	90.0
Sex	male	0.61	1.00	0.43
	female	0.39	0.00	0.57
Educational level	elementary school	0.04	0.14	0.03
	junior high school	0.10	0.54	0.20
	high school	0.36	0.29	0.51
	university	0.50	0.03	0.27
House type	apartment	0.90	0.49	0.74
	independent house	0.10	0.51	0.26
House floor	ground floor or lower	0.44	0.65	0.60
	first or second floor	0.22	0.24	0.16
	third floor or higher	0.34	0.11	0.24
House outer space	balcony	0.30	0.23	0.44
	garden	0.22	0.42	0.40
	none	0.48	0.35	0.17
House window screen	yes	0.84	0.11	0.52
	no	0.16	0.89	0.48
Green area nearby	yes	0.36	1.00	0.74
	no	0.64	0.00	0.26

**Table 2 ijerph-17-03960-t002:** Multivariate odds ratios (ORs) (95% confidence interval) of KAP compared to the Italian group. Model adjusted for age, sex and education level.

		Malayalis	Punjabis
Knowledge	Aedes life cycle	9.04 (6.22, 13.36)	2.49 (1.75, 3.56)
	Aedes bite time	1.6 (1.11, 2.31)	1.24 (0.81, 1.86)
	Aedes transmitted diseases	3.13 (2.15, 4.65)	2.61 (1.03, 6.05)
Attitude	Disease concern	2.39 (1.55, 3.63)	4.98 (2.84, 8.72)
	Aedes nuisance	0.55 (0.38, 0.79)	0.2 (0.13, 0.32)
	Bite nuisance	0.01 (0.0, 0.02)	0.2 (0.13, 0.30)
Practice	Action to avoid Aedes bites	0.73 (0.51, 1.04)	1.56 (1.11, 2.19)

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
