# Peer review of "Knowledge, Attitude and Practices towards the Tiger Mosquito Aedes Albopictus. A Questionnaire Based Survey in Lazio Region (Italy) before the 2017 Chikungunya Outbreak"

_ijerph, 2020, doi:10.3390/ijerph17113960_

Round 1

Reviewer 1 Report

Dear Authors,

This is a particularly useful study as understanding public perceptions, knowledge and associated problems caused by mosquitoes enables scientists and vector control practitioners to design suitable educational but also surveillance and control programmes.

Furthermore, I personally find it really interesting that the participants of your survey are not only Italians but people of Asian (Indian origin) who are usually more robust and tolerant towards mosquitoes and vector borne diseases and that this is also highlighted in your study.

I think one thing that could be improved in this very interesting manuscript is the English language especially in the introductory section. As I am not a native speaker myself, I will try to provide some suggestions but please feel free to double check with a native English speaker. Also I think that some material in the results especially the big long tables could go as supplementary material to allow for the manuscript to be more concise

Please see below my suggestions:

Suggested alternative title: Knowledge, attitude and practices towards the tiger mosquito Aedes albopictus. A questionnaire based survey at the location of the 2017 chikungunya outbreak, Lazio region, Italy.

Abstract

Line 14: outbreak should be in plural tense: outbreaks

Line 15: Add: are important in order to prevent

line 15: Isn't awareness and knowledge almost synonyms? should KAP be knowledge, attitude and practices? in order to agree with your title as well?

line 17: delete d from investigate

Introduction

line 32: please consider rephrasing and also adding a reference: Mosquito -borne disease burden is increasing worlwide following globalization and expansion of travel and trade

line 34: replace naive with new

line 37: consider rephrasing trade of tires and the ability of the tiger mosquito to produce eggs that diapause.....

line 40: delete infected viremia and replace with viremic

line 43: should be communities

line 46: delete the

line 48: what do you mean by non-specific? explain

line 51: Would it be better to put public insteand of patient there

line 52: add the word patients after likelihood of 

line 57: delete represents and replace with is

line 57: delete a before community

line 58: delete s from evidence

line 61: ....have well established popluations but there is a lack of data on the level of knowledge...

line 63: I would rephrase that saying: Additionally there are no data on KAP for the Asian resident communities that may be exposed more due to frequent travelling and contacts in South East Asia where mosquito borne diseases are endemic

line 67: delete parenthesis from Central Italy

Line 69:I would rephrase. The reference group was Italians while the other two groups were Malayalis originally from Kerala, India and Punjabis originally from North India.

Line 69: Since in Indida Aedes transmitted diseases are endemic and frequent outbreaks are recorded (16), a higher level of knowledge and awareness was expected by the Punjabis and Malayalis compared to the Italians.....

Materiald and methods

line 80: replace occurs with occured

line 81: replace manly with mainly

line 85: a convenient sample recruited? do you mean representative sample of participants? 

line 90: replace discarded with rejected

Results

Table 1: should be male insteand of men in the table 

Table 1: consider placing the whole table as supplementary material

line 177: add figure and its number that this info comes from

line 177-179: perhaps they should be placed in the discussion

line 181: Figure 1. Answers regarding tiger mosquito breeding sites....

line 184:  add regarding the before biting time

line187: replace by with but

line 207: add probability value after statitically significant

line 208-211: this whole phrase needs rephrasing as it is imcomprehensible at the moment 

line 226: delete native

line 250: there is a typo in container

line 253: consider changing entomology with entomological

line 255: add that needs before to be filled

line 282:  add the Italian citizens who participated  in the survey instead of the Italian citizens surveyed

line 290: at the time that the questionnaire was administered instead of time of questionnaire administration

line 317: delete carrying replace with more worried

line 321: please consider rephrasing: Comparing the three communities helped highlighting some critical differences in perceptions regarding the coexistence with mosquitoes

line 322: delete s from suggests

line 328: I would add a few lines from lines 305-309 in order to have in my conclusion also some actions that I am suggesting to take

Reviewer 2 Report

MATERIALS AND METHODS

Design:

  • What was the minimum sample needed with respect to the target population? How have you calculated this data? The authors must reflect it.
  • Regarding the question: "What do you do to avoid tiger mosquitoes bites?" You ask an open question and classify it as closed. Why do they do this? Wouldn't it have been better to ask a closed question? Authors must justify their response.

Ethical considerations:

Have you consulted the ethics committee? Authors must mention and say the reference.

REFERENCES

Many bibliographies are obsolete and some citations are incomplete. The bibliographic citations used are more than 5 years old (47, 5%). The authors must update and arrange the bibliography.

Some references do not meet the journal guidelines.

There are references that have errors. Authors should review the citation.

Also, reference 28 is incomplete. This must be corrected.

Round 2

Reviewer 2 Report

Dear authors,

Congratulations on your manuscript.

Best regards